# Complementary Feeding Indicators in Relation to Micronutrient Status of Ghanaian Children Aged 6–23 Months: Results from a National Survey

**DOI:** 10.3390/life11090969

**Published:** 2021-09-15

**Authors:** William E. S. Donkor, Seth Adu-Afarwuah, Rita Wegmüller, Helena Bentil, Nicolai Petry, Fabian Rohner, James P. Wirth

**Affiliations:** 1GroundWork, 7306 Fläsch, Switzerland; rita@groundworkhealth.org (R.W.); nico@groundworkhealth.org (N.P.); fabian@groundworkhealth.org (F.R.); james@groundworkhealth.org (J.P.W.); 2Department of Nutrition and Food Science, University of Ghana, Legon P.O. Box LG 25, Ghana; sadu-afarwuah@ug.edu.gh (S.A.-A.); bentilhelena@gmail.com (H.B.)

**Keywords:** complementary feeding, infant and young child feeding, anemia, iron deficiency, vitamin a deficiency, children

## Abstract

Background: Optimal complementary feeding is critical for adequate growth and development in infants and young children. The associations between complementary feeding and growth have been studied well, but less is known about the relationship between complementary feeding and micronutrient status. Methods: Using data from a national cross-sectional survey conducted in Ghana in 2017, we examined how multiple WHO-recommended complementary feeding indicators relate to anemia and the micronutrient status of children aged 6–23 months. Results: In total, 42%, 38%, and 14% of the children met the criteria for minimum dietary diversity (MDD), minimum meal frequency (MMF), and minimum acceptable diet (MAD), respectively. In addition, 71% and 52% of the children consumed iron-rich foods and vitamin A-rich foods, respectively. The prevalence of anemia, iron deficiency (ID), iron deficiency anemia (IDA) and vitamin A deficiency (VAD) was 46%, 45%, 27%, and 10%, respectively. Inverse associations between MMF and socio-economic status were found, and MMF was associated with an increased risk of ID (55%; *p* < 0.013) and IDA (38%; *p* < 0.002). Conclusion: The pathways connecting complementary feeding and micronutrient status are complex. Findings related to MMF should be further investigated to ensure that complementary feeding programs account for the potential practice of frequent feeding with nutrient-poor foods.

## 1. Introduction

The “1000-day window”—the period between conception and a child’s 2nd birthday—is a key period of growth and development. Due to the benefits and protective effects of exclusive breastfeeding of children <6 months of age that is properly done, growth faltering more frequently occurs in children between 6–23 months of age [1,2], as incidences of non-continued breastfeeding and poor complementary feeding puts infants at risk of becoming malnourished. In low- and low-middle income countries, growth faltering during this period has been extensively documented and attributed to sub-optimal complementary feeding practices [3,4,5] and other risk factors, such as diarrhea, poor sanitary conditions and poor socio-economic status [6,7].

Complementary feeding is the introduction of adequate and nutrient-dense foods to children after 6 months of age, when breastmilk alone cannot meet the energy and nutrient needs of the child [3,5]. This crucial period of transition from exclusive breastfeeding to breastfeeding in tandem with solid and soft foods is a critical window for the promotion of optimal growth, health and development of a child [5]. Prolonged breastfeeding after exclusive breastfeeding in the first 6 months and continued complementary feeding after 6 months increases linear growth [8]. It is known that higher levels of dietary diversity are associated with increased micronutrient density of foods consumed [9,10], and micronutrient deficiencies are strongly associated with childhood morbidity and mortality [11].

In 2006, the World Health Organization (WHO) and other institutions developed a suite of infant and young child feeding (‘IYCF’) indicators [12]. These indicators were designed to assess the status of IYCF practices at the population level, and have been incorporated into national surveys, such as the Demographic and Health Surveys (DHS) and Multiple Indicators Cluster Surveys (MICS) that are implemented in numerous low- and middle-income countries. Of the eight core IYCF indicators, four indicators are related to complementary feeding of children 6–23 months of age [3] and are derived from data collected from a 24 h food frequency dietary questionnaire module. The other core IYCF indicators relate to breastfeeding, or complementary feeding of children in a narrow age range (e.g., children 6–8 months of age). Importantly, the WHO IYCF complementary feeding indicators were not developed using associations with indicators of micronutrient status, such as iron or vitamin A deficiencies. Rather, the WHO IYCF complementary feeding indicators were validated via correlations with indicators of dietary quality and energy intake [13,14]. Nonetheless, there is wide agreement that the WHO IYCF complementary feeding indicators measure feeding practices that are key to child growth and development, and that policymakers can identify challenges to child feeding habits by measuring and tracking these indicators. The WHO IYCF questionnaire tools are used in demographic and health surveys (DHS) and multiple indicator cluster surveys (MICS), the two main population-based surveys implemented throughout the world. The WHO’s IYCF guidelines were updated in 2017, and a revised version of the WHO IYCF guidelines was published in 2021 [15].

Although the WHO IYCF indicators were developed to assess breastfeeding and complementary feeding practices of a population, researchers have previously examined the individual-level associations between IYCF indicators and nutrition, specifically stunting and wasting [16,17], and indicators of chronic and acute malnutrition. Less studied is the association between commonly used complementary feeding indicators and micronutrient status. Obbagy et al. [18] conducted a systematic review of trials and cohort studies, and examined the associations between micronutrient status and the consumption of specific food groups (e.g., meats, cereals, fortified foods). The researchers found that consumption of high-iron foods (e.g., meats, fortified foods) helped to maintain iron stores in children not receiving iron from another source. In an analysis of cross-sectional data, Molla et al. [19] found that unmet minimum dietary diversity and unmet minimum meal frequency were associated with anemia in Ethiopian children 6–23 months of age. Globally, there is scant research examining the relationship between standardized complementary feeding indicators and micronutrient status.

In this study, we thus explore the associations between complementary feeding indicators and demographic factors (i.e., age group, sex, wealth quintile), and examine the linkages between these complementary feeding indicators and anemia, iron deficiency, iron deficiency anemia (IDA), and vitamin A deficiency among children aged 6–23 months in Ghana.

## 2. Materials and Methods

### 2.1. Study Design and Participants

Our analysis utilized data from the Ghana Micronutrient Survey 2017 (GMS 2017), a stratified cross-sectional household-based survey that was conducted between April and June 2017. Ghana’s administrative and agro-ecological zones were used to separate the country into three strata: (1) Southern Belt, consisting mainly of coastal savannah and rainforest and the Western, Central, Great Accra, and Volta regions; (2) Middle Belt, comprising mainly deciduous forest and the Brong Ahafo, Ashanti, and Eastern regions; and (3) Northern Belt, which consists mainly of Guinea/Sudan Savannah and the Northern, Upper West, and Upper East regions.

The survey used a two-stage sampling procedure, with census enumeration areas (EAs) selected using probability proportionate to size and households within each selected EA selected using simple random sampling. Following the random selection of households, the survey recruited all children 6–59 months of age. Further details of the survey design can be found elsewhere [20,21].

### 2.2. Data Collection Procedures

All data were collected by teams consisting of a team leader, interviewers, phlebotomists, and anthropometrists. All field workers received training on the field procedures prior to the field data collection, including classroom-based training and field testing of survey procedures. The performance of all field workers was observed throughout and assessed in a written test to select the best-performing team members.

All teams conducted the household listing exercise and the random selection of households in each EA after which interviewers visited each selected household to administer questionnaires. All the questionnaires were written in English and administered using tablet computers. Questions were translated directly by the interviewer from English into the local language. Interviewers were assigned to different regions according to their language abilities, and the interviews were conducted at the individual level in the language of the participants. Field interviewers administered the household questionnaire (gathering data on household composition, household demographics, socio-economic characteristics, water and sanitation and hygiene and food purchasing patterns) to the household head, or any knowledgeable adult household member in the absence of the head, in each of the selected households.

From the household composition data collected, eligible children (6–59 months of age) and women were automatically selected for the child and woman questionnaires. All children 6–59 months of age were recruited, but women were only recruited from ½ of households. As such, the sample size of mother–child dyads is limited. The woman questionnaire was administered with modules on age, literacy, marital and occupational status, education, dietary diversity, pregnancy and lactation status, antenatal care and consumption of vitamin and mineral supplements. Caregivers of eligible children were interviewed about the child’s age, sex, recent illnesses, infant and young child feeding practices, consumption of micronutrient-rich food and/or supplements and vitamin A supplementation using the child questionnaire. After the completion of all the individual questionnaires, participants were invited to a central place in the EA for blood collection.

Details about the blood collection and processing procedures are described elsewhere [20,21]. A portable hemoglobinometer (Hb301; HemoCue AB, Ångelholm, Sweden) was used to measure hemoglobin concentrations from a capillary blood sample. “Current or recent malaria” infection with multiple malaria parasite species (i.e., *Plasmodium falciparum*, *vivax*, *malariae*, or *ovale*) was determined using a rapid test kit (SD BIOLINE Malaria Ag Pf/Pan; Standard Diagnostics Inc., Giheung-gu, South Korea). As malaria rapid test kits test for specific antigens can take several weeks to clear from the blood [22], they identify both current malaria infections and infections that occurred in the recent past. Concentrations of ferritin, retinol binding protein (RBP), C-reactive protein (CRP), and α1-acidglycoprotein (AGP) were measured at the VitMin laboratory (Willstaett, Germany) using the sandwich ELISA method [23]. Blood pellets were analyzed by the KEMRI-Wellcome Trust Research Programme, (Kilifi, Kenya) to identify sickle cell disease and trait and α-thalassemia status using polymerase chain reactions [24,25,26].

### 2.3. Infant and Young Child Feeding Indicators

Our analysis focuses on five indicators of infant and young child feeding (IYCF). Four of these indicators are included in WHO’s IYCF 2021 guidelines [15]: minimum dietary diversity (MDD), minimum meal frequency (MMF), minimum acceptable diet (MAD), and consumption of iron-rich or iron fortified foods (IRF). Additionally, we created a variable for the consumption of vitamin A rich foods (VARF) for this analysis using the vitamin A-rich food categories in the WHO IYCF guideline’s 24 h recall module.

The WHO IYCF guidelines defined the eight food groups: breastmilk, grains, roots and tubers, legumes and nuts, dairy products (milk, yogurt, cheese), flesh foods (meat, fish, poultry and liver/organ meats), eggs, vitamin-A rich fruits and vegetables, other fruits and vegetables. Plain or fermented corn meal porridge is frequently given to Ghanaian infants and young children, and was added to the “grains” food group. Red palm oil, a commonly-consumed food in Ghana, was also added as a separate food group. In accordance with the WHO IYCF guidelines, red palm oil was not included in the calculation of standard complementary feeding indicators (i.e., indicators A, B, and C, below).

All food consumed by the child in the past 24 h was considered relevant, and there was no minimum quantity of food, except if an item was used as a condiment. Further details of each indicator are given below:Minimum dietary diversity (MDD): Proportion of children 6–23 months of age who receive foods from 5 or more food groups out of eight food groups in the previous 24 h.Minimum meal frequency (MMF): Proportion of children 6–23 months of age who receive solid, semi-solid, or soft foods (but also including milk feeds for non-breastfed children) the minimum number of times or more. The minimum number of times as defined by the WHO for breastfed children is two times for children 6–8 months of age, three times for children 9–23 months of age, and four times for all non-breastfed children 6–23 months of age [15].Minimum acceptable diet (MAD): Proportion of children 6–23 months of age who receive a minimum acceptable diet (apart from breast milk). This is a composite indicator and is defined as the proportion of children 6–23 months of age who achieved both minimum dietary diversity and minimum meal frequency [15].Consumption of iron-rich or iron-fortified foods (IRF): Proportion of children 6–23 months of age who receive an iron-rich food or iron-fortified food that is specially designed for infants and young children, or that is fortified in the home. Iron-rich foods include red meat, poultry, fish and shellfish, organ meats, iron-fortified complementary foods or biscuits, iron-fortified infant formula, and iron-fortified lipid-based nutrient supplements [15].Consumption of vitamin A-rich food, foods fortified with vitamin A (VARF): The proportion of children (6–23 months of age) consuming vitamin A-rich foods or foods fortified with vitamin A. Vitamin A-rich foods include sweet potatoes that are orange or yellow inside, red palm oil, and foods made with red palm oil or palm nuts (“red-red”, palm nut soup, palava sauce: white-seed melon + taro leaves stew).

### 2.4. Data Management and Statistical Analysis

Questionnaire data were directly entered into tablet computers via the Open Data Kit (ODK) software. Hemoglobin concentrations, initially written on paper forms, were transferred into ODK at the end of each day. Blood samples were labeled with unique identifiers matched to each individual, and following the laboratory analysis, results were merged with the child questionnaire data. Data analysis was done using Stata/IC version 14.2.

Using data on each household’s dwelling, water and sanitation conditions and facilities, and ownership of durable goods; a wealth index was calculated by using the method described by the World Bank [27]. The continuous wealth index was categorized into quintiles to enable the cross-tabulation and the subsequent presentation of key indicators by wealth quintile.

The WHO recommendations for anemia were used to define anemia and in preschool children, a cutoff of <110 g/L was set as being anemic. Using the adjustment approach of the Biomarkers Reflecting Inflammation and Nutritional Determinants of Anemia (BRINDA) project, ferritin and RBP concentrations were adjusted for elevated AGP and CRP [28,29]. Iron deficiency in children was defined as inflammation-adjusted serum ferritin < 12 µg/L while vitamin A deficiency in children was defined as RBP < 0.7 µmol/L [30]. Iron deficiency anemia (IDA) was defined as concurrent anemia and iron deficiency in children with both anemia and iron deficiency results.

The statistical precision of all prevalence and mean estimates were assessed using 95% confidence limits, which were calculated accounting for the complex sampling. For dichotomous or categorical data, the chi-square test was applied. To compare the difference between means, the *t*-test was used. Stata’s *nptrend* command was used to identify nonparametric trends in the prevalence of complementary feeding indicators by age group and wealth quintile [31].

Univariate statistics and bivariate associations were weighted to account for the unequal probability of selection of children from the strata used for the GMS 2017. Details of the weighting approach used are described elsewhere [21]. Individual characteristics and the prevalence of feeding indicators and micronutrient status were calculated in aggregate and by child age group. The statistical precision of all prevalence estimates was assessed using 95% confidence limits), accounting for the complex sampling, including the cluster and implicit and explicit stratified sampling used in this survey. For bivariate analyses, *p*-values measuring the differences in prevalence and mean between subgroups were calculated using chi-square and *t*-test, respectively.

Bivariate associations with the key outcome variables (i.e., anemia, ID, IDA, VAD) with *p*-values less than 0.1 were retained for multivariable analyses. A *p*-value < 0.1, rather than 0.05 was preferable for model building to ensure that no potentially important variables were inadvertently excluded from the multivariable analysis [32]. Covariates in the initial model were checked for collinearity by estimating the variance inflation factor (VIF) for all independent variables; a VIF ≥ 4 was used as a threshold for collinearity. After removing collinear variables from the initial models, variables that were not statistically-significant (*p* < 0.05) were removed from the models using backwards elimination.

### 2.5. Ethics and Consent

Ethical approval for the survey protocol was obtained from the Ethics Review Committee of the Ghana Health Service (GHS), number GHS-ERC-15/01/2017. The survey protocol was also registered with the Open Science Framework study registry (doi: 10.17605/OSF.IO/J7BP9) [20]. Verbal informed consent was obtained for household interviews, while written informed consent was given for each child by his/her mother or caregiver. Women also participated following written informed consent. Survey participants who tested positive for malaria or had Hb < 110 g/L were referred to the nearest health facility for further testing and treatment.

## 3. Results

Table 1 describes the general characteristics of children 6–23 months of age. In total, 419 children had valid results for age, sex, residence, strata, and wealth quintile. Similar proportions of male and female children were enrolled in the survey, and a slightly smaller proportion of children were found in the 6–11 months age group compared to the other age groups. Slightly more than half of the children lived in rural households, and almost half of all children resided in the middle belt stratum. Fewer children were in the highest wealth quintile (15.8%) compared to the other quintile groups.

As shown in Figure 1A, approximately 65% of children had been breastfed in the previous 24 h, with markedly higher proportions found in children <18 months of age. About 28% of children aged 6–23 months achieved MDD and 38% achieved MMF, but only 12% of children achieved a MAD. More than 70% and 50% of children consumed IRF or VARF, respectively, in the past day. The proportion of children achieving the six complementary feeding indicators also changed by age group. Statistically significant trends were observed in five out of six indicators. The proportion of children breastfed in the past 24 h had a significant trend (*p* < 0.001) with each age group. The proportion of children consuming IRF and VARF in the previous 24 h significantly (*p* < 0.001) increased by age group. The highest proportion of children were found currently breastfeeding, with more than 88% of children 6–11 months breastfed in the previous 24 h. The proportion of children achieving MMF and MAD decreased with age, but only the decrease of MMF was statistically significant. A significant increasing trend by age was also observed for MDD, although the percentages by age group did not show a steady increase. Further details can be found in Appendix A.

As shown in Figure 1B, the proportion of breastfed children was highest in the lower wealth quintiles. In contrast, higher proportions of children in the highest wealth quintile achieved MAD, MDD and consumed IRF than children in the lower wealth quintiles. Generally, lower proportions of children in the individual wealth categories achieved MAD compared to the other four feeding indicators and this is also seen in the age panel (panel A). Statistically significant trends were observed in four out of six indicators. The proportion of breastfed children decreased, albeit not consistently, as wealth quintile increased. The proportion of children achieving MDD and consuming IRF had a significantly increasing trend as wealth quintile increased (*p* < 0.0001), with markedly more children from the highest wealth quintile achieving MDD. As IRF consists primarily of flesh floods (e.g., fish/seafood, poultry, meats, and organ meats), it is notable that flesh food consumption also increased by wealth quintile, increasing from 51% in the lowest quintile to 89% in the highest quintiles (*p* = 0.02, data not shown). MMF was lowest among children in the highest wealth quintile compared to children in the other wealth categories.

When examining the associations between the complementary feeding indicators and child sex, no statistically significant differences were found.

Table 2 shows foods and food groups consumed in the previous 24 h by age group. The mean number of foods consumed increased from 2.3 in the 6–11-month group to 3.5 in the 18–23-month group. The most common food group fed across all age groups was grains, roots and tubers (90.5%), with a much larger proportion of children consuming foods made from grains (84.7%), such as bread and rice. Breastmilk was the second most consumed food group, with nearly two-thirds of children 6–23 months of age breastfed in the past 24 h. Flesh foods were the third most consumed food group, with fish and seafood being the most commonly consumed across Ghana. About one-tenth of children consumed meat or poultry, and very few children consumed organ meats. Approximately half of all children consumed vitamin A-rich fruits and vegetables, and dark green leafy vegetables (31%) were the most commonly eaten food in this category.

There was a significant and consistent decrease in the proportion of children breastfed in the previous 24 h by age group. The consumption of dairy products varied with age with a lesser proportion of children fed with infant formula and/or sweetened/flavoured milk products. The consumption of infant formula and/or sweetened/flavoured milk products decreased significantly with age. Consumption of eggs and legumes and nuts were generally low compared to the consumption in other major food groups; however, both increased with age respectively. Significant differences in consumption of legumes and nuts, flesh foods, eggs, and vitamin A-rich fruits and vegetables were observed across age groups. Significant differences in consumption of foods in subgroups such as foods made from grains, infant formula and/or sweetened/flavoured milk products, meat and poultry, fish and seafood, vitamin A-rich vegetables, roots and tubers, and vitamin A-rich fruits were observed by age.

The associations between the six complementary feeding indicators and maternal factors, including literacy, employment, and education level were explored (see Appendix A). This exploratory analysis found no significant associations between maternal literacy and any of the six complementary feeding indicators. Maternal employment was only significantly associated with MAD, with a higher proportion of MAD found in children whose mothers had agricultural or unskilled employment. Women’s education was significantly associated with current breastfeeding, MDD, MMF, IRF, and VARF. The proportion of children breastfed or achieving MMF decreased consistently as their mother’s educational level increased. In contrast, the proportions of children achieving MDD, IRF, and VARF were highest among children whose mothers attended senior secondary school or a higher academic institution (e.g., university).

### 3.1. Infant and Young Child Feeding Indicators and Anemia and Micronutrient Status

Table 3 summarizes IYCF indicators and micronutrient status. Almost half of the children 6–23 months had anemia and almost half were iron deficient based on inflammation-adjusted ferritin values. Approximately one-quarter had IDA, and approximately one-tenth of the children were deficient in vitamin A. Regarding MDD, the anemia prevalence was lower among children who achieved MDD compared to those who did not meet the MDD threshold. Similarly, children that had consumed IRF in the previous 24 h had significantly lower anemia prevalence and higher hemoglobin concentrations.

Iron deficiency was positively and significantly associated with MMF and MAD; iron deficiency prevalence was more than 15 and 20 percentage points higher in children meeting the minimum MMF and MAD thresholds. MAD was also significantly associated with ferritin concentration, whereas MMF was not. Only MMF was significantly associated with IDA, with a higher IDA prevalence found in children meeting MMF.

Nearly all associations between vitamin A deficiency and the complementary feeding indicators were not statistically significant. The only association found was between RBP concentration and IRF, with higher concentrations of RBP among children who consumed IRF in the past 24 h.

Table 4 presents the associations between individual food groups and food types/products and anemia, iron deficiency, IDA, and vitamin A deficiency. Significantly lower anemia prevalence were found in children consuming foods made from grains, dairy products, flesh foods, and eggs. Surprisingly, no individual meat group was associated with iron deficiency, but the aggregated food group variable showed that children who consumed meat had a higher iron deficiency prevalence compared to those who did not consume meat (49.5% vs. 37.3%; *p* < 0.043). The legumes and nuts variable showed that children who consumed legumes and nuts had a higher iron deficiency prevalence compared to those who did not consume legumes and nuts (60.0% vs. 41.7%; *p* < 0.023).

Significantly lower IDA prevalence was found in children consuming foods made from grains, roots and tubers and dairy products and in individual food subgroups such as foods made from grains, infant formula, sweetened/flavoured milk products, milk, or yogurt. A higher prevalence of IDA was found among children who consumed legumes and nuts compared to those who did not consume legumes and nuts (37.0% vs. 23.6%; *p* < 0.017). No major food group was significantly associated with VAD. Surprisingly, children who consumed palm oil had a higher vitamin A deficiency prevalence compared to those who did not consume palm oil (17.1% vs. 7.5%; *p* < 0.049).

Other nutrition-related indicators were further explored. Diarrhea in the previous 2 weeks and the prevalence of anemia, iron deficiency, IDA, and vitamin A deficiency; no statistically significant associations were found (data not shown). Appendix A shows the associations between anemia, iron deficiency, IDA, and vitamin A deficiency and anthropometric markers—stunting, wasting, and underweight. No associations were found between anemia and iron deficiency and the aforementioned indicators. IDA was only significantly associated with stunting, with higher levels of IDA found in stunted children. Vitamin A deficiency was significantly associated with stunting and underweight, with a higher proportion of vitamin A deficiency found in both non-stunted and normal weight children.

### 3.2. Determinants of Anemia and Micronutrient Status of Children Aged 6–23 Months

In Table 5, the results of multivariable models of anemia, iron deficiency, IDA, and VAD are presented. In the anemia model, collinearity was observed between two variables: IRF and consumption of flesh foods. As IRF is based on the recent consumption of multiple iron-rich foods (i.e., flesh foods, iron-fortified foods and infant formulas, and lipid-based nutrient supplements containing iron), consumption of flesh foods was excluded as a covariate from the initial anemia model. No collinearity was observed in the initial models for ID, IDA, and VAD.

Consumption of iron-rich foods, eggs in the previous 24 h and a sufficient iron status was less likely linked to anemia. In addition, children with current or recent malaria infection were approximately 70% more likely to be anemic compared to children without malaria.

Current or recent malaria infections was less likely linked to ID and IDA whereas sickle cell trait was less likely linked to ID. Children currently breastfed and consuming legumes and/or nuts had a higher risk of ID and IDA.

A higher wealth quintile and the consumption of grains, roots and tubers was less likely linked to IDA. Children who are male had a higher risk of vitamin A deficiency.

## 4. Discussion

Our analysis showed that the consumption of nearly all food items, apart from infant formula and and/or sweetened/flavoured milk products, increased along with increased child age. While intuitive, this finding illustrates that the common 6–23 month age range used for the analysis of complementary feeding indicators may not always be appropriate. This finding is relevant in Ghana and other countries since identifying optimal and sub-optimal consumption patterns by smaller age groups (e.g., 6–11, 12–17, 18–23 months of age) would be useful to program planners. Furthermore, disaggregation of results by these age groups is recommended in the WHO IYCF guidelines [3]. However, the proportion of children achieving MDD changed only slightly by age group.

Currently breastfed children had a higher risk of iron deficiency and IDA in multivariable analysis. This finding may suggest that breastfed infants and young children are less likely to consume iron-rich foods, and this association has been observed in other studies [33]. However, the direction of this association is not clear. Two studies conducted in Senegal observed an inverse association between malnutrition in young children and prolonged breastfeeding [8,34], yet concluded that prolonged breastfeeding was essentially caused by malnutrition. In brief, the researchers observed that when a child presented symptoms of malnutrition, their mothers would continue breastfeeding, sometimes in place of other foods, in an effort to improve their child’s condition. Two complementary feeding practices were associated with a reduced risk of anemia in our study: consumption of iron-rich foods and the consumption of eggs in the previous 24 h. Consumption of iron-rich foods is likely associated with a decreased risk of anemia by improving children’s iron stores and subsequent production of hemoglobin. However, both consumption of iron-rich foods and iron status were independent predictors of anemia, suggesting that consumption of iron-rich foods may represent other characteristics that could influence anemia status. One such characteristic could be vitamin C intakes from vitamin C-rich fruits that can enhance the bioavailability of non-heme iron leading to improved iron stores, and this possibility is supported by exploratory bivariate analyses showing that a significantly higher proportion of children who consumed iron-rich foods also consumed two food groups that would contain some vitamin-C rich foods: vitamin A-rich foods and vegetables (*p* < 0.01) and other fruits and vegetables (*p* < 0.001).

Children consuming eggs may have a lower risk of anemia as eggs could potentially influence hemoglobin status via increased iron intake. One large egg contains approximately 1.7 mg of iron [35] and thus accounts for 15–44% of a 1- to 3-year-old child’s estimated average requirement of iron [36]. While the mechanism underpinning this association is plausible, our study found no association between egg consumption and ID or IDA. This may suggest that the association between eggs and anemia is potentially due to another nutrient found in abundance in eggs, such as choline, which is converted to phospholipids, which form the outside membrane of red blood cells [37]. While consumption of a nutrient-dense food like eggs could be a proxy of household socioeconomic status, this presumption is refuted by the findings of our preliminary regression models, which showed that egg consumption in the previous 24 h was independently associated with anemia when controlling for household wealth quintile.

The risk of ID and IDA increased in children consuming legumes and/or nuts. While the IYCF questionnaire module did not collect specific information about the type of legumes or nuts consumed, it is likely that most children consumed beans and/or groundnuts as these are staple foods in Ghana and are consumed in considerably higher quantities per capita than any other legumes, nuts, or seeds [38]. Beans and groundnuts both contain large quantities of phytic acid and polyphenols [39], which are some of the most potent inhibitors of non-heme iron absorption [40]. The increased risk of ID and IDA may thus be attributable to the relatively high quantities of those compounds, which have been shown to prevent iron absorption, even at low concentrations [41]. Regarding groundnuts, young children in Ghana are not likely to consume whole groundnuts, but rather groundnut paste (i.e., roasted groundnuts ground into a paste) that is made into groundnut soup or added as a fat source to porridge. While processing strongly reduces the concentration of polyphenols and to a certain extent the concentration of phytic acid [42,43,44,45,46], sufficiently large amounts remain in the foods to reduce iron bioavailability and thus potentially contribute to low iron status.

Although there were associations between malaria and anemia, ID, and IDA, the direction of associations was not consistent. With respect to anemia, recent malaria increased the risk of anemia, and this finding is similar to that found in other studies in some sub-Saharan African countries [47,48,49]. Contrary to anemia, recent malaria was associated with lower risks of ID and IDA. This finding may be reflecting ferritin levels that are elevated after a recent malaria infection due to an acute phase reaction and thus ID prevalence is lower than it would be if the acute phase reaction had already resolved. Similar findings in Ivory Coast showed that CRP and plasma ferritin were elevated even during afebrile malaria; furthermore, it showed a decrease in ID prevalence although it was months after treatment [50]. The study in Ivory Coast was conducted on adolescents, who likely have acquired greater immunity to malaria compared to infants and young children. Thus, acute phase reaction in our study population of children 6–23 months of age may have resulted in a larger increase in ferritin levels that, despite adjustments for inflammation, resulted in artificially lower ID and IDA prevalence. A similar finding between malaria and ID was observed in Sierra Leone [51] and may suggest that iron deficiency in children has a protective effect against malaria infection as low levels of circulating iron in children limit the growth and multiplication of malaria parasites [52] and thus only children with a better iron status would be infected with malaria. However, this finding differs from a recent multi-country study [53] that suggests that malaria contributes to iron deficiency by limiting an individual’s absorption of iron.

Very few associations between VAD and complementary feeding indicators and the consumption of specific foods were found. Although some associations were found between food groups and/or food types and iron deficiency-related outcomes, it is interesting to observe that almost no association with VAD was found.

Lastly, this study illuminates some fundamental features and limitations of composite complementary feeding indicators, particularly. The analysis highlights the fact that the frequently measured complementary feeding indicators are composite indicators. While composite indicators of complementary feeding can be useful for describing the overarching dietary patterns of infants and young children, they can obfuscate the actual dietary patterns. To illustrate, while comprehensive 24 h food frequency data are collected as part of the WHO/UNICEF IYCF questionnaire modules, the data on the consumption of individual foods and food groups are rarely used. Large-scale surveys such as DHS and MICS frequently include all the WHO/UNICEF IYCF questionnaire modules (including the 24 h food frequency questionnaire module), yet only report the results of the composite indicators. In low- and middle-income countries where diets are often monotonous, in particular, a greater exploration and presentation of the consumption of specific foods and food groups can be useful to understanding the specific foods consumed/not consumed.

This study has a few notable limitations. First, as our analysis utilized cross-sectional data, our findings could not identify causal relationships between complementary feeding indicators and anemia or micronutrient deficiencies. Second, the short recall period (i.e., 24 h) used when collecting complementary feeding practices may confound our analysis. Anemia or micronutrient deficiencies are chronic conditions and take weeks or months to develop, and 24 h-recall data may not accurately characterize a child’s routine food consumption and dietary habits. Thirdly, although interviewers were well trained and supervised to correctly translate the questionnaires when conducting interviews in the native language of the respondents, the lack of standard written translations may have led to slightly different translations, and thus interpretations, of questions in the field. That said, due to the uncomplicated nature of questions related to complementary feeding, any bias from lack of formally translated questionnaires is likely to be minimal. Lastly, although our study identifies that dietary pattern changes as children age, our sample size was too small to conduct separate bivariate and multivariable analyses for smaller age groups (e.g., 6–11 m, 12–17 m, 18–23 m).

## 5. Conclusions

The pathways connecting complementary feeding and micronutrient status are complex. Perhaps surprisingly, our study found that currently breastfeeding increased the risk of both ID and IDA. Although the direction of this association is unclear, findings from studies in other countries suggests further investigation is needed to understand this relationship. Our study also found that current or recent malaria reduced the risk of ID and IDA. While not completely explainable, similar results have been observed in other countries and illustrate that there is a complex relationship between malaria infection and iron status. Given the well-documented importance of adequate complementary feeding and the associations we found with anemia, ID, and IDA, programs aimed at improving breastfeeding and complementary feeding practices in Ghana should be strengthened and expanded.

## Figures and Tables

**Figure 1 life-11-00969-f001:**
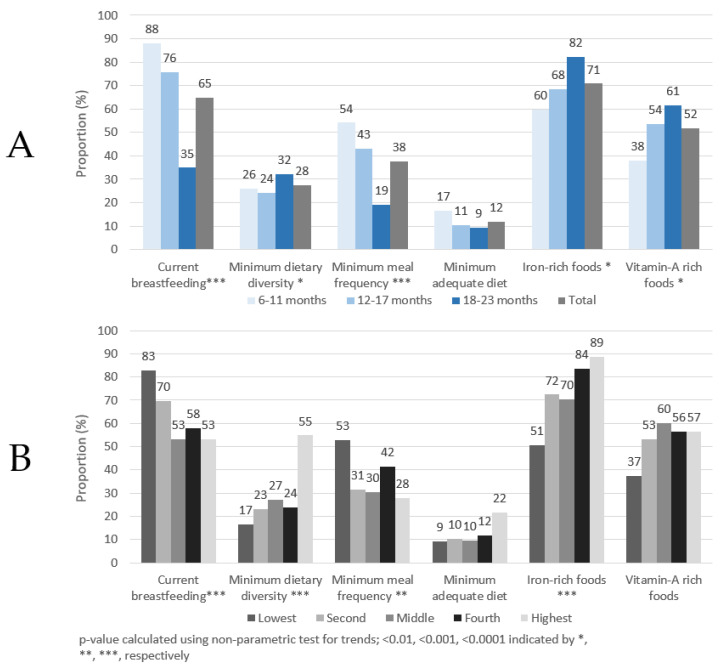
Complementary feeding indicators by age group (**A**) and wealth quintile (**B**), children 6–23 months of age, Ghana, 2017.

**Table 1 life-11-00969-t001:** Characteristics of children aged 6–23 months of age, Ghana, 2017.

Characteristic	*n*	%	(95% CI)
**Age group (in months)**			
6–11 months	127	29.7	(24.4, 35.5)
12–17 months	145	34.7	(29.4, 40.5)
18–23 months	147	35.6	(30.1, 41.5)
**Sex**			
Male	211	50.6	(45.2, 56.1)
Female	208	49.4	(43.9, 54.8)
**Residence**			
Urban	163	45.8	(32.9, 59.3)
Rural	256	54.2	(40.7, 67.1)
**Stratum/belt**			
Southern	111	31.3	(24.2, 39.4)
Middle	178	47.5	(39.8, 55.4)
Northern	130	21.1	(15.8, 27.7)
**Wealth quintile**			
Lowest	144	26.2	(18.5, 35.8)
Second	88	19.3	(13.7, 26.5)
Middle	69	22.4	(15.3, 31.5)
Fourth	61	16.3	(11.5, 22.7)
Highest	57	15.8	(9.6, 24.8)

**Table 2 life-11-00969-t002:** Food groups and food types consumed in the previous 24 h by children 6–23 months of age, Ghana, 2017.

	Food Groups	6–11 Months		12–17 Months		18–23 Months		Total (6–23 Months)		*p*-Value ^c^
		% ^a^	95% CI ^b^	% ^a^	95% CI ^b^	% ^a^	95% CI ^b^	% ^a^	95% CI ^b^	
**1**	**Breast milk**	88.0	(77.5, 93.9)	75.7	(65.6, 83.5)	34.9	(27.7, 43.0)	64.8	(57.8, 71.2)	**<0.001**
**2**	**Grains, roots and tubers**	85.5	(77.8, 90.8)	90.7	(83.9, 94.9)	94.4	(88.6, 97.4)	90.5	(87, 93.1)	0.07
	Thin porridge (e.g., koko)	67.4	(57.3, 76.1)	61.0	(51.2, 69.9)	65.4	(55.7, 73.9)	64.4	(58.5, 69.9)	0.60
	Foods made from grains	74.9	(64.6, 83)	86.4	(79.5, 91.2)	91.1	(84.9, 94.9)	84.7	(80.7, 88)	**0.011**
	White roots and tubers and plantains	16.8	(11, 24.7)	26.6	(19.1, 35.6)	31.4	(24, 39.9)	25.4	(21.3, 30.1)	0.13
**3**	**Legumes and nuts**	10.2	(5.5, 18.1)	13.9	(8.3, 22.4)	26.6	(18.0, 37.4)	17.3	(13.1, 22.7)	**0.01**
**4**	**Dairy products (milk, yogurt, cheese)**	37.9	(28.3, 48.6)	35.7	(24.0, 49.4)	46.9	(35.3, 58.9)	40.4	(32.8, 48.5)	0.30
	Infant formula or sweetened/flavored milk in tins or sachets (liquids module)	15.5	(9.2, 25)	6.8	(3.3, 13.3)	4.4	(1.8, 10.4)	8.5	(5.9, 12.2)	**0.017**
	Milk (tinned, powdered, or fresh animal milk) or Yogurt (liquids)	10.4	(5.9, 17.5)	20.7	(12.3, 32.8)	24.9	(17.1, 34.8)	19.2	(14.6, 24.7)	0.08
	Milk and milk products (food module)	29.7	(20.3, 41.3)	28.3	(18.3, 41.1)	36.7	(26.2, 48.6)	31.7	(24.9, 39.5)	0.55
**5**	**Flesh foods (meat, fish, poultry and liver/organ meats)**	41.8	(30.7, 53.7)	65.0	(56.1, 73)	76.8	(68.0, 83.8)	62.4	(56.2, 68.2)	**<0.001**
	Organ meats	0.2	(0, 1.3)	2.6	(0.7, 8.8)	5.0	(2.3, 10.4)	2.7	(1.4, 5.3)	0.28
	Meat and poultry	1.8	(0.4, 8.3)	10.2	(5.7, 17.3)	22.6	(15.9, 31.1)	12.1	(9.1, 16)	**<0.001**
	Fish and seafood	40.4	(29.7, 52.1)	62.4	(52.8, 71.1)	67.9	(59.3, 75.5)	57.9	(52.1, 63.4)	**<0.001**
**6**	**Eggs**	11.9	(6.6, 20.4)	14.8	(8.9, 23.7)	25.4	(18.6, 33.7)	17.7	(13.5, 23)	**0.02**
**7**	**Vitamin A rich fruits and vegetables**	37.9	(28.5, 48.4)	53.6	(43.2, 63.7)	61.4	(52.5, 69.5)	51.8	(45.5, 57.9)	**0.002**
	Vitamin A-rich vegetables, roots and tubers	1.8	(0.5, 6.4)	11.8	(7.5, 18.2)	3.4	(1.3, 8.5)	5.8	(3.8, 8.9)	**0.011**
	Dark green leafy vegetables	22.8	(14.4, 34.1)	31.1	(22.8, 40.9)	37.5	(29.1, 46.7)	31.0	(24.9, 37.7)	0.41
	Vitamin A-rich fruits	6.6	(3.1, 13.6)	11.3	(6.2, 19.6)	24.1	(16.9, 33.1)	14.5	(10.4, 19.8)	**0.032**
**8**	**Other fruits and vegetables**	18.1	(12.5, 25.6)	16.1	(10.9, 23)	27.0	(19.2, 36.5)	20.6	(16.7, 25.2)	0.08
**9**	**Red palm oil**	24.1	(16.5, 33.9)	27.7	(18, 40.1)	28.6	(21.1, 37.5)	27.0	(21.7, 33)	0.91
**Mean dietary diversity score**		2.3	(2.0, 2.7)	2.8	(2.6, 3.1)	3.5	(3.2, 3.8)	2.9	(2.7, 3.1)	**<0.001**
**Proportion of children with minimum dietary diversity**		25.9	(16.4, 38.5)	24.2	(16.4, 34.2)	32.2	(23.7, 42.2)	27.6	(22.2, 33.7)	0.46

^a^ Percentages weighted for unequal probability of selection. ^b^ CI = confidence interval, calculated taking into account the complex sampling design. ^c^ *p*-values measuring the differences in prevalence between the age groups using chi-square test.

**Table 3 life-11-00969-t003:** Prevalence of anemia, iron deficiency, iron deficiency anemia, vitamin A deficiency by complementary feeding indicator, Ghana, 2017.

	% Anemia (*n* = 398) ^a,c^	*p*-Value ^b^	Mean Hb (g/L) ^a^	*p*-Value ^b^	% Iron Def. (*n* = 398) ^a,c^	*p*-Value ^b^	Mean Ferritin (ug/L) ^a^	*p*-Value ^b^	% IDA ^a,c^	*p*-Value ^b^	% Vit A Def. (*n* = 398) ^a,c^	*p*-Value ^b^	Mean RBP (µmol/L) ^a^	*p*-Value ^b^
**Minimum Dietary Diversity**														
Yes	34.7	0.027	111.8	0.025	44.5	0.89	18.0	0.62	19.3	0.12	8.1	0.49	1.09	0.22
No	49.1		108.0		45.2		19.1		28.3		11.2		1.05	
**Minimum Meal Frequency**														
Yes	48.2	0.47	108.5	0.63	54.8	0.013	17.3	0.23	37.5	0.002	9.3	0.64	1.05	0.72
No	43.7		109.5		39.5		19.8		20.3		11.0		1.07	
**Minimum Acceptable Diet**														
Yes	43.3	0.82	109.0	0.77	67.0	0.023	13.9	0.017	33.4	0.36	12.5	0.82	1.03	0.53
No	45.6		109.8		42.4		19.5		25.1		10.1		1.06	
**Consumption of iron rich food, foods fortified with iron**														
Yes	38.3	<0.001	112.0	<0.001	46.1	0.45	19.9	0.63	24.9	0.41	9.4	0.38	1.09	0.015
No	61.5		104.5		41.9		18.9		30.0		12.6		0.99	
**Consumption of vitamin-A rich food, foods fortified with vitamin A**														
Yes	40.0	0.07	109.4	0.61	44.9	0.99	18.2	0.54	22.2	0.24	12.0	0.96	1.06	0.86
No	50.7		108.5		44.9		19.5		30.9		8.7		1.07	
**TOTAL**	**45.7**	**-**	**108.8**	**-**	**45.1**	**-**	**18.7**	**-**	**26.7**		**10.3**	**-**	**1.06**	**-**

^a^ Percentages/means weighted for unequal probability of selection. ^b^ *p*-values measuring the differences in prevalence and mean between subgroups were calculated using chi-square and *t*-test, respectively. ^c^ Anemia defined as hemoglobin <120 g/L; iron deficiency (ID) is defined as BRINDA inflammation-adjusted serum ferritin <12 µg/L; iron deficiency anemia (IDA) is defined as concurrent anemia and ID. Vitamin A deficiency defined as BRINDA inflammation-adjusted serum retinol binding protein <0.7 µmol/L.

**Table 4 life-11-00969-t004:** Associations between food groups and food types and anemia, iron deficiency, IDA, and vitamin A deficiency, children aged 6–23 months, Ghana 2017.

	Food Groups	Anemia	ID	IDA	VAD
**1**	**Breastmilk**	O	** ↑ **	** ↑ **	O
**2**	**Grains, roots and tubers**	** ↓ **	O	** ↓ **	O
	Thin porridge (e.g., koko)	** ↓ **	O	O	O
	Foods made from grains	** ↓ **	O	** ↓ **	O
	White roots and tubers and plantains	O	O	O	O
**3**	**Legumes and/or nuts**	O	** ↑ **	** ↑ **	O
**4**	**Dairy products (milk, yogurt, cheese)**	** ↓ **	O	** ↓ **	O
	Infant formula and/or sweetened/flavored milk products in tins or sachets (liquids module)	O	O	** ↓ **	O
	Milk (tinned, powdered, or fresh animal milk) or Yogurt (liquids module)	O	O	** ↓ **	O
	Milk and milk products (food module)	** ↓ **	O	O	O
**5**	**Flesh foods (meat, fish, poultry and liver/organ meats)**	** ↓ **	** ↑ **	O	O
	Organ meats	O	O	O	O
	Meat and poultry	O	O	O	O
	Fish and seafood	** ↓ **	O	O	O
**6**	**Eggs**	** ↓ **	O	O	O
**7**	**Vitamin A rich fruits and vegetables**	O	O	O	O
	Vitamin A-rich vegetables, roots and tubers	O	O	O	O
	Dark green leafy vegetables	O	O	O	O
	Vitamin A-rich fruits	O	O	O	O
**8**	**Other fruits and vegetables**	O	O	O	O
**9**	**Red palm oil**	O	O	O	** ↑ **

Note: This table presents bivariate associations between consumption of various foods and outcome indicators. Associations indicated when chi-square *p*-value is less than 0.1. **↓** = consumption of food significantly associated with lower prevalence of anemia/ID/IDA/VAD; **↑** = consumption of food significantly associated with higher prevalence of anemia/ID/IDA/VAD; O = no significant difference.

**Table 5 life-11-00969-t005:** Adjusted relative risk of anemia, iron deficiency, iron deficiency anemia, and vitamin A deficiency in children aged 6–23 months, Ghana 2017.

Model	Characteristic	Category	Adjusted Relative Risk ^a^	95% CI
** *Anemia model (n = 372)* ^b^ **	Consumed iron-rich foods in past 24 h	Yes	0.61	(0.50, 0.75)
		No	referent	-
	Consumed eggs in past 24 h	Yes	0.54	(0.37, 0.80)
		No	referent	-
	Iron status ^b^	Sufficient	0.52	(0.42, 0.65)
		Deficient	referent	-
	Malaria status ^c^	Positive	1.74	(1.36, 2.21)
		Negative	referent	-
** *Iron deficiency model (n = 362)* ^b^ **	Currently breastfeeding	Yes	1.58	(1.21, 2.05)
		No	referent	-
	Consumed legumes and/or nuts in past 24 h	Yes	1.26	(1.03, 1.56)
		No	referent	
	Malaria status ^c^	Positive	0.26	(0.14, 0.50)
		Negative	referent	-
	Sickle cell status ^d^	HbAS, HbSS	0.68	(0.47, 0.98)
		Normal	referent	-
	Child Sex	Male	1.27	(1.05, 1.56)
		Female	referent	-
** *Iron deficiency anemia model (n = 362)* ^b^ **	Currently breastfeeding	Yes	1.53	(1.03, 2.31)
		No	referent	-
	Consumed grains, roots and tubers in past 24 h	Yes	0.66	(0.44, 0.99)
		No	referent	
	Consumed legumes and/or nuts in past 24 h	Yes	1.43	(1.02, 2.02)
		No	referent	
	Malaria status ^c^	Positive	0.38	(0.19, 0.74)
		Negative	referent	-
	Wealth quintile	First	referent	
		Second	0.81	(0.55, 1.20)
		Middle	0.59	(0.36, 0.99)
		Fourth	0.63	(0.38, 1.05)
		Highest	0.43	(0.22, 0.82)
** *Vitamin A deficiency model (n = 398)* **	Child Sex	Male	1.92	(1.01, 3.65)
		Female	referent	-

Note: All regression models contain child age in months as a continuous variable. ^a^ Adjusted relative risk calculated using Poisson regression. ^b^ Anemia defined as hemoglobin <120 g/L; iron deficiency (ID) is defined as BRINDA inflammation-adjusted serum ferritin <12 µg/L; iron deficiency anemia (IDA) is defined as concurrent anemia and ID. Iron deficiency defined as BRINDA inflammation-adjusted serum ferritin concentrations <12 µg/L. ^c^ Malaria positive defined as current or recent malaria infection by at least on Plasmodium species (i.e., falciparum, vivax, malariae, or ovale). ^d^ Sickle cell disease and trait represented by HbSS and HbAS, respectively.

## Data Availability

The data underlying the results presented in the study are owned by UNICEF Ghana and the Ministry of Health Ghana and contain confidential, identifying information. Data are available from UNICEF Ghana (accra@unicef.org) for researchers who meet the criteria for access to confidential data. The authors had no special access to the data.

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
