# Peer review of "Complementary Feeding Indicators in Relation to Micronutrient Status of Ghanaian Children Aged 6–23 Months: Results from a National Survey"

_life, 2021, doi:10.3390/life11090969_

Round 1
Reviewer 1 Report
Dear Authors, you presented a national study from Ghana showing associations between IYCF practices and micronutrient status.
In general, the study has been presented well. However, I do have two concerns in regard to your methodology and consequently to your results and discussions.
1: you used the former MDD indicator using 4 out 7 pre-defined food groups. This indicator has been updated to include breast milk intake as first food group summing up to 8 food groups with a threshold of 5 out of 8 food groups. I propose you check on whether the reanalysis with the new indicator can be done or that you at least discuss potential implications of this in the discussion section.
2. following up on the first comment - I am missing a profound inclusion of breast milk intake information. We are talking about the complementary feeding age, thus, breast milk intake is considered as important contributor to the health of the children. Please include breast milk intake into the MDD models and/or micronutrient models depending on which indicator you apply in the end to ensure that you do not misinterprete your findings.
More detailed comments have been inlcuded into the document.
Reviewer 2 Report
This is an interesting study evaluating the complementary feeding indicators in Ghanian children aged 6 to 23 months. I do have few questions and suggestions for the authors.
Introduction: Please revise the manuscript for grammatical errors. Line 38 and 39, I believe the risk factors are poor sanitary conditions and lower socio economic status.
Methods: In addition to the socio economic status, maternal literacy level is crucial in a Child's nutritional status and the type of food made at home. Do authors have data on the literacy level? Can it be used as a separate variable in the analysis?
Do authors have access to the anthropometric measurements of the subjects to correlate with the feeding practices. As the authors have mentioned this is a cross sectional data and so correlating with anthropometric data would be more meaningful if data is available.
Do authors have data on the recent history of acute gastroenteritis? As AGE is a major comorbidity in developing countries leading to malnutrition under 5 and affect dietary intake?
Although there is an inverse relationship reported between MMF and IDA/ID, I believe there are multiple confounders which needs to be addressed before arriving at any particular conclusion.
Round 2
Reviewer 1 Report
Dear Authors, the manuscript has substantially improved.
The comments previously made have been adequately adressed apart the one on vitamin C. see line 340ff. It is well noticed that there is no specific food group on vitamin C, however, fruits are quite likely to be high in vitamin C. You could either include a comment on this topic in the discussion that vitamin C intake influences the bioavailabilty of iron absorption or you consider it as limitation not beeing able to control for vitamin C intake.
A second limitation which should be included is the lack of standard translations of the questionnaire which may lead to different understandings of the questions in the field which may result in biased data.
Line 537ff: the sentence seems incomplete.
